# Gene regulatory network inference using mixed-norms regularized multivariate model with covariance selection

**Alain J. Mbebi**[1,2], **Zoran Nikoloski**[1,2]*

**1** Bioinformatics Department, Institute of Biochemistry and Biology, University of Potsdam, Karl-Liebknecht-Str. 24-25, Germany, **2** Systems Biology and Mathematical Modeling Group, Max Planck Institute of Molecular Plant Physiology, Am Mühlenberg 1, Germany

* nikoloski@mpimp-golm.mpg.de

**Data Availability Statement:** The approaches are implemented using the R programming language and the codes are freely available from https://github.com/alainmbebi/mixed-norms-GRN. All data underlying this publication are publicly available

## Abstract

Despite extensive research efforts, reconstruction of gene regulatory networks (GRNs) from transcriptomics data remains a pressing challenge in systems biology. While non-linear approaches for reconstruction of GRNs show improved performance over simpler alternatives, we do not yet have understanding if joint modelling of multiple target genes may improve performance, even under linearity assumptions. To address this problem, we propose two novel approaches that cast the GRN reconstruction problem as a blend between regularized multivariate regression and graphical models that combine the $L_{2,1}$-norm with classical regularization techniques. We used data and networks from the DREAM5 challenge to show that the proposed models provide consistently good performance in comparison to contenders whose performance varies with data sets from simulation and experiments from model unicellular organisms *Escherichia coli* and *Saccharomyces cerevisiae*. Since the models' formulation facilitates the prediction of master regulators, we also used the resulting findings to identify master regulators over all data sets as well as their plasticity across different environments. Our results demonstrate that the identified master regulators are in line with experimental evidence from the model bacterium *E. coli*. Together, our study demonstrates that simultaneous modelling of several target genes results in improved inference of GRNs and can be used as an alternative in different applications.

## Author summary

Reconstruction of cellular networks based on snapshots of molecular profiles of the network components has been one of the key challenges in systems biology. In the context of reconstruction of gene regulatory networks (GRNs), this problem translates into inferring regulatory relationships between transcription factor coding genes and their targets based on, often small, number of expression profiles. While unsupervised nonlinear machine learning approaches have shown better performance than regularized linear regression approaches, the existing modeling strategies usually do predictions of regulators for one target gene at a time. Here, we ask if and to what extent the joint modeling of regulation

and their corresponding references provided in the article. All used data are also provided on the indicated GitHub.

**Funding:** AJM and ZN are supported by the European Union's Horizon 2020 research and innovation programme in connection with the projects BREEDCAFS [GA No. 727934] https://www.breedcafs.eu/ and PlantaSYST [FPA No. 664620] https://plantasyst.eu/. The funders had no role in study design, data collection and analysis, decision to publish, or preparation of the manuscript.

**Competing interests:** The authors have declared that no competing interests exist.

for multiple targets leads to improvement of the accuracy of the inferred GRNs. To address this question, we proposed, implemented, and compared the performance of models cast as a blend between regularized multivariate regression and graphical models that combine the $L_{2,1}$-norm with classical regularization techniques. Our results demonstrate that the proposed models, despite relying on linearity assumptions, show consistently good performance in comparison to existing, widely used alternatives.

## Introduction

Elucidation of gene-regulatory networks (GRNs), comprising the entirety of transcription factor (TF)-target gene interactions, remains one of the key challenges in systems biology studies of single cells and entire organisms [1]. Advances in technologies for probing gene-regulatory interactions, including: Chromatin immunoprecipitation combined with sequencing (ChIP-Seq) [2], Yeast one hybrid (Y1H) [3], and DNA-affinity purification sequencing (DAP-Seq) [4], have facilitated understandings in the *in vivo* and *in vitro* binding of TFs to the promoter region of target gene and have provided valuable resources for obtaining insights in the characteristics of GRNs across organisms [5, 6]. However, these technologies are still resource-intensive even when applied with model organisms. As a result, addressing this key challenge of systems biology necessitates the development of computational approaches for reconstruction of GRNs that rely on other data sources, such as gene expression, that capture in part the effect of TF binding and subsequent activation or repression of transcription of the target gene.

The computational approaches for GRN reconstruction use data from steady-state and/or time-resolved experiments; they rely on unsupervised, semi-supervised, and supervised machine learning methods [7–9] to identify TFs that explain the expression (patterns) of target genes (TGs). Recent advances in supervised learning of GRNs have benefited from the compendia of TF-target gene interactions obtained by the aforementioned technologies [10]. Irrespective of the data used and the machine learning approach applied, reconstruction of GRNs is often performed with considerably fewer observations ($n$) than number of predictors ($p$) that has resulted in the development and application of diverse regularization techniques in Gaussian graphical models (GGMs) [11, 12] and the regression setting [13–15]. Further, due to the often non-linear dependence between the expression of TGs and their regulating TFs, machine learning techniques based on random forests [16–18] and kernels in combination with regressions [19] have resulted in improved accuracy of GRN reconstruction with data from *Escherichia Coli* and *Saccharomyces cerevisiae* [20].

Computational approaches for GRN reconstruction from gene expression data in the regression setting model the expression of each TG based on the expression of the TFs as predictors. In doing so, the relation between TGs is neglected in the process of model building [21]. Therefore, it remains unexplored if the simultaneous consideration of multiple TGs in the linear setting may perform as well as the models for individual targets in a non-linear setting.

Evidence from analysis of existing GRNs have demonstrated the presence of master regulators [22], *i.e.* TFs that regulate a sizeable proportion of target genes. The existing approaches either reconstruct GRNs assuming a prior that given TFs act as master regulators [23] or infer master regulators from the models built for the individual target genes. Furthermore, ChIPseq data have demonstrated the dependence of gene regulatory interactions on the biological context, determined by the interaction among the environment, developmental stage, and cell type/tissue [24]. Therefore, gene regulatory interactions are plastic and this characteristic is

often neglected in the reconstruction of GRNs, particularly with data from multiple environmental perturbations and/or organisms, resulting in the reconstruction of consensus interactions [14].

To tackle these shortcomings, we propose two novel GRN reconstruction approaches as a blend between regularized multivariate regression and graphical models in the large-$p$-small-$n$ setting. Specifically, by assuming that the observed gene expression data matrix is drawn from a multivariate normal distribution, we impose the $L_{2,1}$-norm penalty on the regression coefficients along with the $L_1$ (or $L_2$) on the precision matrix to jointly model the gene expression of all TGs in the penalized likelihood framework. While the $L_{2,1}$-norm has been previously used for identification of gene network module [25] and representative genes [26], these approaches do not explicitly address the problem of GRN reconstruction, and when they do [27], prior information about the number of regulators is required. In the current work, we leverage the $L_{2,1}$-norm's feature selection ability and show that model formulation allows us to use an iterative scheme in which the estimate of the precision matrix is used to refine the regression coefficient estimates at the next iteration until convergence. Using gene expression data sets from *E. coli* and *S. cerevisiae* as well as *in silico* data from the Dialogue on Reverse Engineering Assessment and Methods (DREAM5) network inference challenges [20], we evaluate the performance of the proposed models via extensive comparative analyses with respect to the state-of-the-art methods and show the advantages of the proposed approaches in addressing the two mentioned shortcomings–the identification of master regulators and the detection of plastic interactions.

## Results and discussion

### Preliminaries and notation

Before presenting the models, which represents one of our results, we introduce the notation used in the rest of the manuscript. Let $m^i$ and $m_j$ be respectively the $i^{\text{th}}$ row and $j^{\text{th}}$ column of a matrix $\mathbf{M} = (m_{ij})$. $\mathbf{M}^{-1}$ and $\mathbf{M}^{\mathbf{T}}$ represent respectively, the inverse and the transpose of $\mathbf{M}$. $\mathbf{I}_n$ stands for the $n$-dimensional identity matrix, and if $m_i$ is the $i^{\text{th}}$ component of the vector $m \in \mathbb{R}^n$, then its $L_p$-norm is defined as

$$\left\| m \right\|_p = \left( \sum_{i=1}^n \left\| m_i \right\|^p \right)^{\frac{1}{p}}. \tag{1}$$

The $L_{2,1}$-norm [28] of a matrix $\mathbf{M} \in \mathbb{R}^{k \times l}$ and its partial derivative with respect to $\mathbf{M}$ are respectively

$$\left\| \mathbf{M} \right\|_{2,1} = \sum_{i=1}^k \sqrt{\sum_{j=1}^l m_{ij}^2} = \sum_{i=1}^k \left\| m^i \right\|_2 \tag{2}$$

and $\frac{\partial}{\partial \mathbf{M}} \left\| \mathbf{M} \right\|_{2,1} = 2\mathbf{QM}$, where $\mathbf{Q} \in \mathbb{R}^{k \times k}$ is the diagonal matrix with entries $q_{ii} = \frac{1}{2\|m^i\|_2}$.

In the regression setting for GRN inference, we aim to quantify the regulatory relationship between $s$ TGs (i.e. response variables) $y_1, \cdots, y_s$ and a single set of $p$ TFs (i.e. predictor variables) $x_1, \cdots, x_p$, such that $y_k = b_{1k}x_1 + \cdots + b_{pk}x_p + \varepsilon_k$, $1 \le k \le s$. The model can then be cast in the matrix notation as

$$\mathbf{Y} = \mathbf{XB} + \mathbf{E}, \tag{3}$$

where $\mathbf{Y}_{n \times s} = (\boldsymbol{y_1}, \cdots, \boldsymbol{y_n})^T$, $\mathbf{X}_{n \times p} = (\boldsymbol{x_1}, \cdots, \boldsymbol{x_n})^T$, $\mathbf{B}_{p \times s} = (\boldsymbol{b_1}, \cdots, \boldsymbol{b_p})^T$ and $\mathbf{E}_{n \times s} = (\boldsymbol{\varepsilon_1}, \cdots, \boldsymbol{\varepsilon_n})^T$ are respectively the TGs (i.e. response), TFs (i.e. predictors), regulatory links (i.e. regression coefficients) and error matrices.

Assuming that the errors $\boldsymbol{\varepsilon_i}$ are independent and normally distributed with covariance matrix $\boldsymbol{\Sigma}$ (i.e. $\boldsymbol{\varepsilon_i} \overset{i.i.d}{\sim} \mathcal{N}_s(0, \boldsymbol{\Sigma})$), then the negative log-likelihood function [29] of the parameters $(\mathbf{B}, \boldsymbol{\Omega})$ can be written up to a constant as

$$\mathcal{L}(\mathbf{B}, \boldsymbol{\Omega}) = \mathrm{Tr}\left[\frac{1}{n}(\mathbf{Y} - \mathbf{XB})^T(\mathbf{Y} - \mathbf{XB})\boldsymbol{\Omega}\right] - \log|\boldsymbol{\Omega}|, \qquad (4)$$

where $\boldsymbol{\Omega} = \boldsymbol{\Sigma}^{-1}$ is the precision matrix, Tr denotes the trace linear operator and $|\boldsymbol{\Omega}|$ is the determinant of the matrix $\boldsymbol{\Omega}$. Estimators of the parameters $\mathbf{B}$ and $\boldsymbol{\Omega}$ derived from standard procedures such as maximum likelihood and weighted least-squares are equivalent to those obtained when regressing each of the $s$ responses on the $p$ predictors separately. However, these estimators have poor performances, are computationally unstable and less efficient for prediction when the number of predictor and response variables are larger than the sample size.

As noted above, existing regression-based approaches for GRN reconstruction neglect the correlation among the response variables (*i.e.* TGs). To address this issue, we construct new sparse estimators for the regression coefficient and precision matrix via penalized likelihood optimization. Specifically, for tuning parameters $\lambda_1 \geq 0$, $\lambda_2 \geq 0$ and by penalizing the negative log-likelihood in Eq (4), the $s(s + 1)/2$ parameters of the precision matrix $\boldsymbol{\Omega}$ are used to update the estimate of the regression coefficient $\mathbf{B}$ at the next iteration until convergence. In the following, we provide estimates $\widehat{\mathbf{B}}$ and $\widehat{\boldsymbol{\Omega}}$ as solution to the mixed $L_1L_{2,1}$-norms and $L_2L_{2,1}$-norms regularized multivariate regression and covariance selection problems. For clarity, the terms experiment, condition and time point are used interchangeably; and mixed-norms terminology in this context simply refers to the fact that, the $L_1$ (or $L_2$) and $L_{2,1}$ penalties are simultaneously imposed on $\boldsymbol{\Omega}$ and $\mathbf{B}$ in the proposed optimization problems.

## Mixed $L_1L_{2,1}$-norms regularized multivariate regression and covariance selection

When the constant term with no effect on the optimization over $\mathbf{B}$ and $\boldsymbol{\Omega}$ is ignored, the objective function to be minimized for the mixed $L_1L_{2,1}$-norms is proportional to

$$\mathcal{L}_1(\mathbf{B}, \boldsymbol{\Omega}) = \underset{(\mathbf{B}, \boldsymbol{\Omega})}{\mathrm{argmin}}\, \mathrm{Tr}\left[\frac{1}{n}(\mathbf{Y} - \mathbf{XB})^T(\mathbf{Y} - \mathbf{XB})\boldsymbol{\Omega}\right]$$

$$- \log|\boldsymbol{\Omega}| + \lambda_1 \sum_{i \neq j}|\omega_{ij}| + \lambda_2 \left\|\mathbf{B}^T\right\|_{2,1}. \qquad (5)$$

Notice how the $L_{2,1}$ penalty is imposed on $\mathbf{B}^T$ instead of $\mathbf{B} \in \mathbb{R}^{p \times s}$, since: (i) we work under the usual assumption that the number of TF genes ($p$) is considerably smaller than the number of TGs ($s$), (ii) each TF is likely to regulate many TGs [30], and (iii) the $L_{2,1}$ penalty may push some entries in $\mathbf{B}$ (i.e. TF-TG interaction) toward zero. As a result, this formulation facilitates model interpretation and the identification of candidate for interactions and master TFs. The latter can be seen by looking closely to Eq (2) and realizing that the $L_1$-norm encourage simultaneously row sparsity in $\mathbf{B}^T$ whereby, the $i$th predictor's effect is quantified with the $L_2$-norm, while summation over all data points is achieved with the $L_1$-norm. This motivate the choice of $L_{2,1}$-norm regularization. The optimization problem in Eq (5) is biconvex. Therefore, convexity is ensured when solving for either parameter $\mathbf{B}$ or $\boldsymbol{\Omega}$, while keeping the other fixed.

Solving for $\mathbf{B}$ with $\Omega$ fixed to $\Omega_0$, Eq (5) reduces to the convex:

$$\widehat{\mathbf{B}}(\Omega_0) = \underset{\mathbf{B}}{\operatorname{argmin}} \operatorname{Tr}\left[\frac{1}{n}(\mathbf{Y} - \mathbf{XB})^T(\mathbf{Y} - \mathbf{XB})\Omega_0\right] + \lambda_2\left\|\mathbf{B}^T\right\|_{2,1}. \tag{6}$$

Taking the partial derivative with respect to $\mathbf{B}$ yields

$$\frac{\partial\mathcal{L}_1(\mathbf{B}, \Omega_0)}{\partial\mathbf{B}} = -\frac{2}{n}\mathbf{X}^T(\mathbf{Y} - \mathbf{XB})\Omega_0 + 2\lambda_2\mathbf{BC}, \tag{7}$$

where $\mathbf{C}$ is the diagonal matrix with the $i^{\text{th}}$ diagonal entry $c_{ii} = 1/(2\|\mathbf{b}_i\|_2)$. For computational stability, one can also use $c_{ii} = 1/\left(2\sqrt{(\mathbf{b}^i)^T(\mathbf{b}^i) + \zeta}\right)$ as an approximation [31], with $\zeta \to 0$.

**Solving the mixed $L_1L_{2,1}$-norms model for $\mathbf{B}$.** The first-order condition defined by Eq (7) gives the following inohomogeneous Sylvester equation [32] in term of $\mathbf{B}$:

$$\mathbf{X}^T\mathbf{XB} + n\lambda_2\mathbf{BC}\Omega_0^{-1} = \mathbf{X}^T\mathbf{Y}. \tag{8}$$

Using Kronecker product and the vec operator [33], one can rewrite Eq (8) as the following ($sp \times sp$) linear system $\left[\mathbf{I}_s \otimes (\mathbf{X}^T\mathbf{X}) + (n\lambda_2\mathbf{C}\Omega_0^{-1})^T \otimes \mathbf{I}_p\right]\operatorname{vec}(\widehat{\mathbf{B}}) = \operatorname{vec}\left[\mathbf{X}^T\mathbf{Y}\right]$, that is more facile to solve. However, for gene expression data, $s$ is often too large such that attempting to solve Eq (8) using this transformation becomes computationally prohibitive due to high memory requirements. We address this limitation (see Method 1 in S1 Text for details), by using the singular value decomposition (SVD) of $\mathbf{X} = \mathbf{U}_1\Gamma_1\mathbf{V}_1^T$, the matrix inversion lemma [34] and change of variables in Eq (9)

$$\begin{cases} \tilde{\mathbf{B}} & = \mathbf{V}_1^T\mathbf{B} \in \mathbb{R}^{n\times s} \\ \mathbf{S} & = \mathbf{V}_1^T\mathbf{X}^T\mathbf{Y} \in \mathbb{R}^{n\times s} \\ \mathbf{K} & = \mathbf{C}\Omega_0^{-1} \in \mathbb{R}^{s\times s} \\ \Gamma_1^T\Gamma_1 & = \operatorname{diag}(\gamma_1, \gamma_2, \cdots, \gamma_n) \in \mathbb{R}^{n\times n} \end{cases} \tag{9}$$

to obtain $\mathbf{B} = \mathbf{V}_1\tilde{\mathbf{B}} \in \mathbb{R}^{p\times s}$. We refer to the latter as the $L_1L_{2,1}$- solution. Notice that, the proposed estimate can be viewed as a generalization of several existing approaches. Of special interest in our comparative analysis is the special case when the diagonal matrix $\mathbf{C} = \mathbf{I}_s$. Under this assumption, the $L_{2,1}$-norm regularization on the regression coefficient matrix becomes $\operatorname{Tr}(\mathbf{B^TB})$, and the optimization problem becomes the multi-output regression [35] with identity task covariance. It is interesting to point out that, the regularization $\operatorname{Tr}(\mathbf{B^TB})$ is equivalent to imposing a Gaussian prior on $(\mathbf{B^TB})^{1/2}$. Herein, this particular estimate is referred to as $L_1L_{2,1}$G-solution. For details on other special cases such as the $L_{2,1}$ feature selection [31], the ridge and the ordinary least square as well as explanations regarding their derivation, we refer the reader to Method 2 in S1 Text.

**Solving the mixed $L_1L_{2,1}$-norms model for $\Omega$.** For fixed $\mathbf{B}$ at a chosen point $\mathbf{B}_0$ and when solving for $\Omega$, the optimization problem in Eq (5) yields

$$\widehat{\Omega}(\mathbf{B}_0) = \underset{\Omega}{\operatorname{argmin}}\left[\frac{1}{n}(\mathbf{Y} - \mathbf{XB}_0)^T(\mathbf{Y} - \mathbf{XB}_0)\Omega\right] + \lambda_1\sum_{i\neq j}|\omega_{ij}|. \tag{10}$$

This corresponds to the $L_1$-penalized covariance estimation problem and the graphical LASSO [36] (GLASSO) can be used to derive $\Omega$ for the model in Eq (10).

## Mixed $L_2L_{2,1}$-norms regularized multivariate regression and covariance selection

Analogously to the optimization problem in Eq (5), we formulate the following mixed $L_2L_{2,1}$-norms objective function:

$$\mathcal{L}_2(\mathbf{B}, \boldsymbol{\Omega}) = \underset{\mathbf{B}, \boldsymbol{\Omega}}{\operatorname{argmin}} \operatorname{Tr} \left[ \frac{1}{n} (\mathbf{Y} - \mathbf{XB})^T (\mathbf{Y} - \mathbf{XB}) \boldsymbol{\Omega} \right]$$

$$- \log |\boldsymbol{\Omega}| + \lambda_1 \left\| \boldsymbol{\Omega} \right\|_2 + \lambda_2 \left\| \mathbf{B}^T \right\|_{2,1}. \tag{11}$$

**Solving the mixed $L_2L_{2,1}$-norms model for B.** When solving for **B** with fixed $\Omega$, the proposed mixed $L_2L_{2,1}$-norms model in Eq (11) which imposes the $L_2$ penalty on $\Omega$ yields similar solutions as the optimization problem in Eq (6). Using similar methodology as S1 Method in S1 Text, we obtain the $L_2L_{2,1}$ and $L_2L_{2,1}$G-solutions, for respectively the main problem and the special case (*i.e.* when a Gaussian prior is imposed on $(\mathbf{B^T B})^{1/2}$).

**Solving the mixed $L_2L_{2,1}$-norms model for $\Omega$.** For fixed **B** at a chosen point $\mathbf{B}_0$ the optimization problem in Eq (11) when solving for $\Omega$ becomes

$$\widehat{\boldsymbol{\Omega}}(\mathbf{B}_0) = \underset{\boldsymbol{\Omega}}{\operatorname{argmin}} \operatorname{Tr} \left[ \frac{1}{n} (\mathbf{Y} - \mathbf{XB}_0)^T (\mathbf{Y} - \mathbf{XB}_0) \boldsymbol{\Omega} \right]$$

$$- \log |\boldsymbol{\Omega}| + \lambda_1 \left\| \boldsymbol{\Omega} \right\|_2, \tag{12}$$

where the partial derivative with respect to $\Omega$ is given by

$$\frac{\partial \mathcal{L}_2(\mathbf{B}_0, \boldsymbol{\Omega})}{\partial \boldsymbol{\Omega}} = \frac{1}{n} (\mathbf{Y} - \mathbf{XB}_0)^T (\mathbf{Y} - \mathbf{XB}_0) - \boldsymbol{\Omega}^{-1} + 2\lambda_1 \boldsymbol{\Omega}. \tag{13}$$

Defining $\mathbf{P} = \frac{1}{n} (\mathbf{Y} - \mathbf{XB}_0)^T (\mathbf{Y} - \mathbf{XB}_0)$ and setting Eq (13) to zero, we obtain the following quadratic matrix equation:

$$2\lambda_1 \boldsymbol{\Omega}^2 + \mathbf{P} \boldsymbol{\Omega} - \mathbf{I}_s = 0 \tag{14}$$

which is a special form of the well known algebraic Riccati equation encountered in multiple fields such as control theory and optimization [37, 38]. However, because the fundamental theorem of algebra is not valid for matrix polynomials, problems in the form of Eq (14) are often difficult to solve even in the matrix square root case $\mathbf{X}^2 = \mathbf{A}$ [39]. Therefore we ask if our problem then has a solution, which we answer by the affirmative (cf. Method 3 in S1 Text) and show that the solution to our problem exists and is uniquely given by

$$\boldsymbol{\Omega}(\mathbf{B}_0) = \frac{1}{2\lambda_1} \left[ (\mathbf{P}^2 + 8\lambda_1 \mathbf{I}_s)^{\frac{1}{2}} - \mathbf{P} \right]. \tag{15}$$

## Remark: Existence and uniqueness of a positive definite solution for quadratic matrix equations

It is known that equations of the form $\mathbf{AX}^2 + \mathbf{BX} + \mathbf{C} = 0$, $\quad \mathbf{A}, \mathbf{B}, \mathbf{C} \in \mathbb{R}^{s \times s}$ can have no solution, a finite positive number or infinitely many solutions [40], but to the best of our knowledge, we found no particular evidence regarding the existence and uniqueness of solutions. However, while solving Eq (14) we noticed that, if $\mathbf{A} = \mathbf{I}_s$, **B** and **C** commute and are

respectively non-negative and positive definite, and if $\mathbf{B}^2 - 4\mathbf{C}$ is positive definite, then the existence and uniqueness of a positive definite solution $\mathbf{X}$ is guarantied and can be explicitly determined using the usual formula of the roots in the scalar case. With the positive definiteness requirement of the covariance and correlation matrices [11] being a major drawback in the situation where the sample size $n$, is smaller than the number of variables $s$ (e.g. for microarray data sets), this existence and uniqueness of a positive definite solution can be of particular relevance when using GGM for GRN reverse engineering.

## Comparative analysis with DREAM5 data sets

The performance of the proposed inference approaches (i.e. $L_1L_{2,1}$ and $L_2L_{2,1}$ along with their variants $L_1L_{2,1}G$ and $L_2L_{2,1}G$) are compared with that of GENIE3, TIGRESS, ANOVerence, PLSNET, ENNET, PORTIA, etePORTIA and D3GRN when reconstructing the regulatory networks of *E. coli*, *S. cerevisiae* and the simulated data (i.e. *in silico*) with similar regulatory dynamic as *E. coli*. The contending methods are chosen to include the winner of the challenge (i.e. GENIE3, TIGRESS and ANOVerence), as well as some of the most recent state-of-the-art approaches (i.e. PLSNET, ENNET, PORTIA, etePORTIA and D3GRN) applied on the same data sets. For network-specific assessment and in contrast to all evaluated methods which exhibit large variability in performance across networks, Table 1 and Fig 1, show that the proposed models show consistently good performance across all data sets. Overall and as depicted in the last three columns of Table 1 and S1 Table, the proposed approaches have comparable performances to that exhibited by the best method. Specifically, in terms of AUROC and

**Table 1. Comparison of model performance using area under the ROC curve (AUROC) and area under the precision-recall curve (AUPR) on DREAM5 data sets.**

| Methods | *In silico* (Network 1) | | *E. coli* (Network 3) | | *S. cerevisiae* (Network 4) | | Score | | |
|---|---|---|---|---|---|---|---|---|---|
| | AUROC | AUPR | AUROC | AUPR | AUROC | AUPR | AUROC | AUPR | Overall |
| L1L21 | 0.800 | 0.264 | 0.633 **0.670**\* | 0.105 0.108\* | 0.538 | 0.023 | 0.648 0.660\* | 0.086 0.086\* | 0.367 **0.373**\* |
| L1L21G | 0.800 | 0.254 | 0.640 **0.670**\* | 0.109 0.106\* | **0.539** | 0.023 | 0.651 **0.661**\* | 0.086 0.085\* | 0.368 **0.373**\* |
| L2L21 | 0.800 | 0.264 | 0.633 **0.670**\* | 0.104 0.108\* | 0.538 | 0.023 | 0.648 0.660\* | 0.085 0.086\* | 0.367 **0.373**\* |
| L2L21G | 0.800 | 0.254 | 0.640 **0.670**\* | 0.109 0.107\* | **0.539** | 0.023 | 0.651 **0.661**\* | 0.086 0.085\* | 0.368 **0.373**\* |
| GENIE3 | 0.811 | 0.285 | 0.616 | 0.093 | 0.517 | 0.020 | 0.636 | 0.080 | 0.358 |
| ANOVerence | 0.778 | 0.247 | 0.662 | **0.111** | 0.519 | 0.021 | 0.644 | 0.083 | 0.363 |
| TIGRESS | 0.778 | 0.295 | 0.593 | 0.068 | 0.516 | 0.020 | 0.619 | 0.073 | 0.346 |
| PLSNET | **0.847** | 0.232 | 0.569 | 0.044 | 0.514 | 0.020 | 0.628 | 0.058 | 0.343 |
| ENNET | 0.791 | **0.408** | 0.571 | 0.048 | 0.502 | 0.018 | 0.609 | 0.070 | 0.340 |
| PORTIA | 0.813 | 0.352 | 0.619 | 0.101 | 0.536 | **0.027** | 0.646 | 0.098 | 0.372 |
| etePORTIA | 0.815 | 0.356 | 0.619 | 0.102 | 0.536 | **0.027** | 0.646 | **0.099** | 0.372 |
| D3GRN | 0.780 | 0.354 | 0.653 | 0.081 | **0.539** | 0.021 | 0.649 | 0.084 | 0.367 |

Performances of the proposed $L_1L_{2,1}$ and $L_2L_{2,1}$ along with their variants $L_1L_{2,1}G$ and $L_2L_{2,1}G$ based on the optimal regularization parameters obtain with 10–folds CV, are compared with that of the winner of the challenge (i.e. GENIE3, ANOVerence, TIGRESS) and some of the most recent state-of-the-art approaches (i.e. PLSNET, ENNET, PORTIA, etePORTIA and D3GRN). The last three columns include scores used to quantify the overall assessment of all inference approaches across the three networks under investigation. The star symbol is used to indicate that the corresponding value is obtained with a diagonal estimated precision matrix (i.e. with the proposed approaches), entries in bold represent the best performance for each column and we used the R package "precrec" to compute the AUROC and AUPR with default parameters for each algorithm.

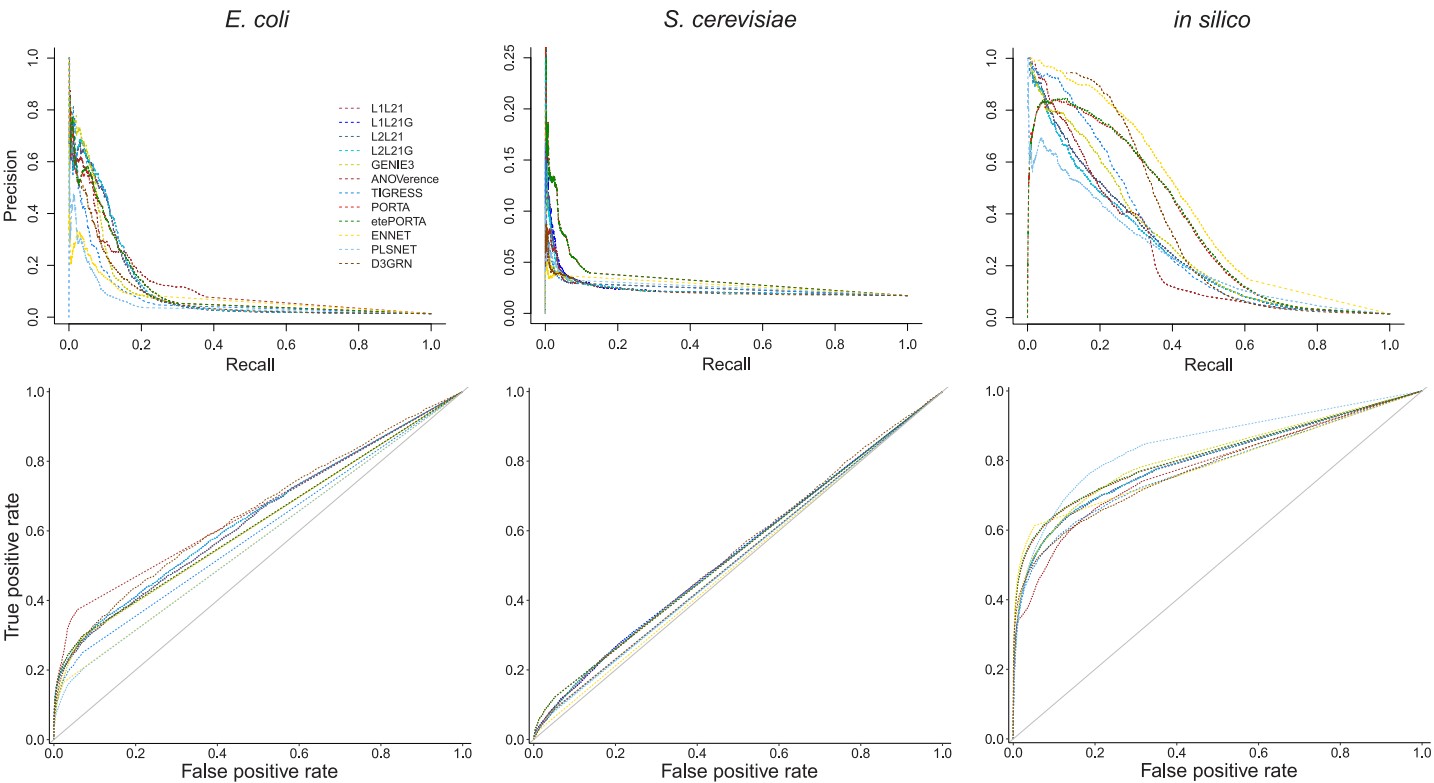

**Fig 1. PR and ROC curves for individual methods in the comparative analysis with DREAM5 data sets.** We used the $L_1L_{2,1}$, $L_2L_{2,1}$, their respective variants (*i.e.* $L_1L_{2,1}G$, and $L_2L_{2,1}G$), the winner of the challenge (i.e. GENIE3, ANOVerence and TIGRESS), and some of the most recent state-of-the-art approaches (i.e. PLSNET, ENNET, PORTIA, etePORTIA and D3GRN) to infer the regulatory networks of *E. coli* (left), *S. cerevisiae* (middle) and *in silico* (right). Shown in the upper and lower panels are respectively the precision-recall (PR) and receiver operating characteristic (ROC) curves.

Overall scores, the proposed models slightly outperform the contenders while the best performing state-of-the-art method (i.e. etePORTIA) in the comparative analysis shows an improved AUPR score of 1.3% compared to the former. With consistent performances across all evaluated data sets, we conclude that the proposed models are competitive and reliable alternatives to state-of-the-art GRN inference methods.

## Comparative analysis with LCL data sets

Results summarized in Table 2 show that, except for TIGRESS (AUROC = 0.510) at the individual network level on Geuvadis for lymphoblastoid cell lines, the highest peformance is always achieved by one of the proposed approaches for all considered metrics and data sets. Despite an improved performance exhibited by the proposed methods compared to the contenders that were also considered in a recent comparative analysis [41] on the same data sets, we reach a similar conclusion (*i.e.* for AUROC and AUPR), whereby all models exhibit relatively low performance that can be attributed to the complexity of *in vivo* networks and high sparsity of the ground truth used for evaluation. Because of their performance consistency across all considered networks and their ability for early detection of true positive edges (*i.e.* nCDG), and the fraction of true positive in the top-$k$ predictions (*i.e.* EP) ($k$ is the number of true positive in the gold standard), we conclude that the proposed approaches are competing alternative for GRN inference.

**Table 2. Comparison of model performance using area under the ROC curve (AUROC), area under the precision-recall curve (AUPR), early precision (EP) and normalized discounted cumulative gain (nDCG) on LCL data sets.**

| Methods | LCL (Geuvadis) | | | | LCL (Niu) | | | | Score | | |
|---|---|---|---|---|---|---|---|---|---|---|---|
| | AUROC | AUPR | EP (%) | nDCG | AUROC | AUPR | EP (%) | nDCG | AUROC | AUPR | Overall |
| L1L21 | 0.507 | 0.139 | **99.84** | 0.347 | 0.518 | 0.145 | 64.75 | 0.359 | 0.512 | 0.141 | 0.327 |
| L1L21G | 0.502 | **0.145** | 99.51 | **0.359** | **0.528** | **0.153** | 65.51 | **0.361** | **0.514** | **0.148** | **0.331** |
| L2L21 | 0.507 | 0.139 | **99.84** | 0.347 | 0.518 | 0.145 | 64.75 | 0.359 | 0.512 | 0.141 | 0.327 |
| L2L21G | 0.502 | **0.145** | 99.51 | **0.359** | 0.518 | 0.144 | **99.70** | 0.345 | 0.509 | 0.144 | 0.327 |
| GENIE3 | 0.498 | 0.135 | 21.47 | 0.000 | 0.494 | 0.132 | 27.90 | 0.214 | 0.495 | 0.133 | 0.314 |
| TIGRESS | **0.510** | 0.144 | 78.57 | 0.224 | 0.501 | 0.137 | 89.93 | 0.000 | 0.505 | 0.140 | 0.322 |
| PLSNET | 0.483 | 0.126 | 37.11 | 0.000 | 0.494 | 0.132 | 30.41 | 0.240 | 0.488 | 0.128 | 0.308 |
| ENNET | 0.490 | 0.131 | 76.25 | 0.301 | 0.489 | 0.130 | 67.25 | 0.273 | 0.489 | 0.130 | 0.309 |
| PORTIA | 0.504 | 0.139 | 71.09 | 0.173 | 0.502 | 0.138 | 75.41 | 0.280 | 0.502 | 0.138 | 0.320 |
| etePORTIA | 0.504 | 0.139 | 71.02 | 0.148 | 0.503 | 0.138 | 75.61 | 0.284 | 0.503 | 0.138 | 0.320 |
| D3GRN | 0.498 | 0.135 | 19.62 | 0.169 | 0.499 | 0.135 | 20.50 | 0.000 | 0.498 | 0.135 | 0.316 |

Performances of the proposed $L_1L_{2,1}$ and $L_2L_{2,1}$ along with their variants $L_1L_{2,1}G$ and $L_2L_{2,1}G$ based on the optimal regularization parameters obtain with 10−folds CV are compared with that of GENIE3, TIGRESS, PLSNET, ENNET, PORTIA, etePORTIA and D3GRN. The last three columns include scores used to quantify the overall assessment of all inference approaches across Geuvadis and Niu expression data sets. Entries in bold represent the best performance for each column and we used the R package "precrec" to compute the AUROC and AUPR with default parameters for each algorithm.

## Analysis with *E. coli* data across multiple conditions

Gene regulation depends on the cellular context including the cell type and the environmental conditions [42]. In this section, we focus on the latter and study master TFs involved in the regulatory dynamic of *E. coli* across multiple stress conditions. To this end, we applied our proposed models with data comprising few time-resolved samples gathered from *E. coli* strain MG1655 exposed to cold, heat, lactose-diauxic shift and oxidative stress conditions.

A previous comparative analysis with the same data contrasted the performance of the fused LASSO extension [14] with nine state-of-the-art inference methods. After re-evaluating all the models we reached a similar conclusion, whereby the fused LASSO achieves better performance and assigns higher scores to the true regulatory links. For this reason, we used the fused LASSO model as a benchmark when assessing the performance of our proposed approaches. Following the same methodology for performance assessment and for a fair comparison, a combination of TFs in RegulonDB [43] and DREAM5 challenge was considered, to finally obtain 173 TFs and 1561 TGs for GRN inference. Our findings summarized in Table 3 show that, one of the proposed inference methods generally achieved the highest performance with respect to AUROC and AUPR, except on the oxidative stress condition where fused LASSO exhibited the highest AUROC. Despite the small improvement shown by the proposed approaches, overall all method achieved relatively low AUROC and AUPR on this data sets. This could be explained in part, by the imbalanced structure of the gold standard and the very small sample size. As expected, and consistent with previous studies [44, 45], the results on combined data sets show an improved performance for all inference approaches with respect to AUROC and AUPR, as the sample size increased (i.e. from 5 for each condition to 20 for the combined data sets). Recalling the caveats for using AUROC and AUPR to compare inference methods with different level of sparsity, further assessment using EP and nCDG reported in Table 3 show the superiority of the proposed methods.

Next, using the master regulator identification's procedure (see Materials and methods) and considering all TFs interacting with more than 50% (i.e. $\alpha$) TGs, we compiled in S2 Table,

**Table 3. Comparison of model performance using AUROC, AUPR, EP and nDCG on time-resolved transcriptomics data sets for model organism *E. coli*.**

| Methods | | L1L21 | L1L21G | L2L21 | L2L21G | Fused LASSO | GENIE3 | PLSNET | ENNET | PORTIA | etePORTIA | D3GRN |
|---|---|---|---|---|---|---|---|---|---|---|---|---|
| **Cold** | AUROC | 0.531 | **0.534** | 0.529 | 0.530 | 0.502 | 0.494 | 0.487 | 0.492 | 0.528 | 0.526 | 0.468 |
| | AUPR | **0.015** | **0.015** | **0.015** | **0.015** | 0.014 | 0.013 | 0.012 | 0.013 | **0.015** | 0.014 | 0.011 |
| | EP (%) | 49.03 | 48.64 | 50.04 | **52.71** | 49.10 | 38.16 | 22.24 | 31.67 | 40.12 | 40.30 | 20.13 |
| | nDCG | **0.162** | 0.155 | 0.155 | 0.155 | 0.142 | 0.120 | 0.150 | 0.136 | 0.155 | 0.152 | 0.110 |
| **Heat** | AUROC | 0.515 | 0.515 | 0.515 | 0.507 | 0.490 | 0.480 | 0.510 | 0.495 | **0.517** | 0.514 | 0.497 |
| | AUPR | **0.014** | **0.014** | **0.014** | **0.014** | 0.012 | 0.010 | 0.012 | 0.012 | **0.014** | 0.013 | 0.012 |
| | EP (%) | 50.19 | 49.46 | 50.19 | **50.53** | 43.80 | 26.80 | 23.43 | 40.14 | 38.30 | 39.11 | 20.30 |
| | nDCG | 0.282 | 0.302 | 0.282 | **0.394** | 0.236 | 0.156 | 0.257 | 0.182 | 0.292 | 0.322 | 0.162 |
| **Oxidative** | AUROC | 0.521 | 0.502 | 0.521 | 0.502 | **0.532** | 0.510 | 0.503 | 0.507 | 0.522 | 0.520 | 0.511 |
| | AUPR | **0.015** | 0.013 | **0.015** | 0.013 | **0.015** | 0.014 | 0.014 | 0.012 | **0.015** | **0.015** | 0.010 |
| | EP (%) | 45.88 | 46.07 | 45.88 | 46.07 | **54.17** | 12.17 | 23.40 | 13.41 | 33.40 | 36.43 | 13.69 |
| | nDCG | **0.177** | 0.170 | **0.177** | 0.170 | 0.167 | 0.154 | 0.124 | 0.141 | 0.167 | 0.168 | 0.117 |
| **Lactose** | AUROC | 0.501 | **0.504** | 0.501 | **0.504** | 0.502 | 0.502 | 0.499 | 0.486 | 0.504 | **0.507** | 0.496 |
| | AUPR | 0.013 | **0.014** | 0.013 | **0.014** | 0.013 | 0.013 | **0.014** | **0.014** | **0.014** | 0.013 | 0.010 |
| | EP (%) | 51.93 | **51.98** | 51.93 | **51.98** | 48.16 | 28.16 | 12.26 | 22.32 | 34.27 | 32.22 | 9.28 |
| | nDCG | 0.160 | 0.131 | 0.160 | 0.131 | 0.140 | 0.171 | 0.130 | 0.092 | 0.221 | **0.224** | 0.113 |
| **Combined data sets** | AUROC | **0.564** | 0.561 | **0.564** | 0.561 | 0.556 | 0.539 | 0.559 | 0.543 | 0.550 | 0.540 | 0.522 |
| | AUPR | **0.017** | **0.017** | **0.017** | **0.017** | **0.017** | **0.017** | 0.015 | 0.017 | **0.017** | **0.017** | 0.016 |
| | EP (%) | **56.63** | 55.51 | **56.63** | 55.51 | 38.32 | 48.78 | 22.40 | 42.40 | 32.80 | 32.41 | 17.20 |
| | nDCG | **0.148** | 0.128 | **0.148** | 0.128 | 0.142 | 0.146 | 0.000 | 0.123 | 0.136 | 0.140 | 0.039 |
| **Score** | AUROC | 0.524 | 0.523 | **0.525** | 0.520 | 0.508 | 0.505 | 0.511 | 0.504 | 0.524 | 0.521 | 0.498 |
| | AUPR | **0.014** | **0.014** | 0.014 | 0.014 | 0.013 | 0.013 | 0.013 | 0.013 | **0.015** | 0.014 | 0.012 |
| | Overall | 0.269 | 0.268 | **0.270** | 0.267 | 0.261 | 0.259 | 0.262 | 0.259 | 0.269 | 0.268 | 0.255 |

The performances of the proposed methods are contrasted with that of Fused LASSO, GENIE3, TIGRESS, PLSNET, ENNET, PORTIA, etePORTIA and D3GRN under four experimental conditions including heat, cold, lactose, oxidative as well as their combination. Scores in the last three columns are also shown to quantify the overall performance of the inference approaches across all data sets. Recalling that with the same data sets, the fused LASSO was already assessed and outperformed the contending approaches, the current comparative analysis implicitly extends to Gaussian graphical models (GGM), the algorithm for the reconstruction of accurate cellular networks(ARACNE), GENIE3, global silencing, CLR and LASSO-type (i.e. $L_1$, $L_0$ and $L_{1/2}$) regularization. Entries in bold represent the best performance, and AUROC and AUPR were computed using the R package "precrec".

the list of MR[1] conserved across all conditions. Although originally designed for gene tissue specificity, we adapted the $\tau$-index [46] as shown in Eq (16) to compute condition specificity of MR[1] and MR[2] that we previously identified to be conserved across conditions

$$\tau = \frac{\sum_{i=1}^{n}(1 - \widehat{x}_i)}{n - 1}; \quad \widehat{x}_i = \frac{x_i}{\max_{1 \leq i \leq n(x_i)}} \tag{16}$$

Here, $n$ is the number of conditions, $x_i$ the gene expression in the $i^{\text{th}}$ condition and $\widehat{x}_i$ the normalized (*i.e.* by the maximal component value) expression profile. It can be observed that $\tau \in [0, 1]$ and depending on the obtained value, we infer that the corresponding master TF is a housekeeping gene (i.e. $\tau \to 0$) or condition-specific (i.e. $\tau \to 1$). Following [46] and [47], that respectively considered $\tau \geq .85$ and $.8$ as a threshold for tissue specificity, we used as decision rule ($\tau > .8$) to check if the given master regulator is ubiquitously expressed or not. Interestingly our finding is in agreement with the $\tau$-index (cf. S2 Table), whereby all MR[1] that the proposed $L_1L_{2,1}$ and $L_2L_{2,1}$ found conserved across all four conditions have their specificity index

below the threshold of 0.8. Using the derived $\tau$-index as a sanity check, we conclude that these master transcription factors are indeed conserved across all conditions.

In contrast, $MR^2$ are only found conserved across three of the four stress conditions (i.e. cold, lactose and oxidative). This is in line with the study by [48] in which it was suggested that *E. coli* perceives high temperatures as a sign of inflammation, and as a result downregulates flagella class II and III genes (to avoid detection by the host immune system). This process is caused by the lower level of upstream activator *flhD* that we found conserved under other three stress conditions. Additionally, the presence of *flhD* and *flhC* in our list of conserved master regulator is quite interesting as these have been previously identified as master regulator for the expression of flagellar genes in *E. coli* [49, 50]. Similarly, the absence of conservation of the transcription factor *CspA* under heat condition could be justified, since it is among the major cold shock proteins of *E. coli* [51] that are only induced upon temperature decrease. Specifically, it has been shown that the induction of *CspA* is mainly caused by dramatic stabilization of its mRNA at low temperature [52, 53].

The study of sparsity level in our estimated regression coefficients and precision matrix shows that the expression of 1,156 genes was under the regulation of all 173 TFs used for the analysis (i.e. none of the rows of regression coefficients or in the precision matrix was entirely zero). Cold was the stress condition for which the three $MR^1$, *fliZ*, *alaS* and *fis*, regulated respectively about 57%, 56% and 53% of the 1,156 genes (cf. S2 Table). In contrast, lactose was the stress for which the $MR^2$ regulated the smallest number of TGs (cf. S2 Table). To further investigate if the conserved $MR^1$ and $MR^2$ share any biological attributes, we performed enrichment analysis using the web application "ShinyGO" [54] while correcting for multiple testing with false discovery rate (FDR) (p-value $< 0.05$). The enrichment analysis (GO biological process) reveals that the conserved $MR^1$ (cf. S1A Fig) are mostly enriched for negative regulation of RNA biosynthesis process, nucleic acid-templated transcription and nucleobase-containing compound metabolic process. Moreover, $MR^2$ (cf. S1B Fig) conserved under cold, lactose and oxidative stress conditions are mostly enriched in three biological processes including regulation of organelle, bacterial-type flagellum and cell projection assembly.

## Conserved $MR^2$ across tumour and normal tissues from NSCLC exhibit low SEG−index suggesting their housekeeping nature

In this section, we further assess the ability of the proposed $L_1L_{2,1}$ and $L_2L_{2,1}$ to identify master regulators conserved across different conditions (*i.e.* tumour and healthy). To this end, we analyzed a large expression profile data set comprising 10077 genes from 1118 non-small cell lung cancer tissue samples of which 925 are affected by squamous cell carcinoma, adenocarcinoma and large cell carcinoma tumour, and 193 correspond to clinically healthy. After identifying the top-$k$ $MR^2$ in each type, we interrogated their intersection to find those conserved across tumour and normal states. For better readability, we sought to mention that for the NSCLC data set at hand, we considered $k = 26$ because below this value, all $MR^2$ in normal condition identified by the proposed method had less than 3% (*i.e.* about 164) regulatory links with the corresponding TGs. As shown in Table 4, we found that $MR^2$ in tumour samples exhibit the highest connection with the associated target genes. In addition, the study of their intersection in tumour and normal samples identified *CXXC5, ZBED1, PPARA, PBX3, SREBF1, FOXC1* and *ARNT2* to be conserved across both types. Because of the involvement of housekeeping genes in basic cell maintenance, their expression levels is expected to be constant regardless of their specific roles, cell types or experimental conditions [55, 56]. Therefore, we asked if the list of our $MR^2$ found conserved in both tissues could be categorized as housekeeping genes or not. For this purpose, we used the stably expressed gene index (SEG) [57] as further validation

**Table 4. Identified tissue specific MR$^2$, their associated SEG−index and respective proportion of links with target genes in the inferred network.**

| MR$^2$ | Normal % of links | SEG−index | MR$^2$ | Tumour % of links | SEG−index |
|---|---|---|---|---|---|
| *HMGN3* | 3.169 | 0.542 | *MXI1* | 10.638 | 0.439 |
| *SREBF2* | 3.042 | 0.609 | ***CXXC5*** | 10.018 | **0.407** |
| *RBPJ* | 3.206 | 0.687 | ***ZBED1*** | 10 | **0.329** |
| ***CXXC5*** | 3.88 | **0.407** | ***PPARA*** | 10.583 | **0.414** |
| *ZNF395* | 3.26 | 0.562 | *ZHX2* | 10.984 | 0.699 |
| ***ZBED1*** | 3.005 | **0.329** | *ZNF32* | 9.982 | 0.523 |
| ***PPARA*** | 3.77 | **0.414** | *TEAD2* | 10.237 | 0.672 |
| *FAM200B* | 3.388 | 0.707 | *MGA* | 10.036 | 0.708 |
| ***PBX3*** | 3.497 | **0.422** | *ZNF503* | 10.237 | 0.416 |
| *SMAD3* | 3.534 | 0.539 | ***PBX3*** | 10.182 | **0.422** |
| *NR1H3* | 3.297 | 0.528 | *SMAD1* | 9.964 | 0.567 |
| *DEAF1* | 3.297 | 0.597 | ***SREBF1*** | 10.073 | **0.301** |
| ***SREBF1*** | 3.26 | **0.301** | *DDIT3* | 9.964 | 0.597 |
| *MECOM* | 3.005 | 0.506 | *TRERF1* | 10.401 | 0.439 |
| *HEY1* | 3.224 | 0.355 | ***FOXC1*** | 10.164 | **0.408** |
| *CEBPA* | 3.406 | 0.56 | *OSR2* | 10.073 | 0.513 |
| *GLIS3* | 3.388 | 0.462 | *NFE2L3* | 10.036 | 0.523 |
| *FOXQ1* | 3.552 | 0.307 | ***ARNT2*** | 10.874 | **0.427** |
| *TFCP2L1* | 3.06 | 0.439 | *FOXP2* | 10.237 | 0.317 |
| ***FOXC1*** | 3.188 | **0.408** | *ESR1* | 9.927 | 0.462 |
| *L3MBTL4* | 3.06 | 0.357 | *PLAG1* | 9.927 | 0.324 |
| ***ARNT2*** | 3.315 | **0.427** | *ASCL2* | 10.437 | 0.271 |
| *MYB* | 3.206 | 0.529 | *AHRR* | 10.036 | 0.295 |
| *SP5* | 3.097 | 0.41 | *NKX3−1* | 10.31 | 0.411 |
| *IRX1* | 3.224 | 0.336 | *MYCN* | 10.601 | 0.436 |
| *NR0B1* | 3.133 | 0.492 | *ISL1* | 10.146 | 0.44 |

Using the proposed L$_1$L$_{2,1}$ and L$_2$L$_{2,1}$, we derived a list of type 2 master transcription factors genes (i.e. MR$^2$) in tumour and normal tissues for NSCLC data sets. Genes in bold represent the conserved MR$^2$ across both tissues type. Also reported is the percentage of links each identified MR$^2$ has with the target genes along with the stably expressed genes (SEG) index that is a metric characterizing housekeeping genes at the single cell level.

step. Interestingly, the SEG−index of all conserved MR$^2$ are less than 0.5 suggesting their housekeeping nature is in line with our result. Theoretically, one should expect MR$^2$ to exhibit the lowest SEG−index. However, most definitions of housekeeping genes do not account for alternative splicing, whereby a gene can stably expresses different transcripts in diverse tissues or cells [58, 59]. As a matter of fact and as shown in Table 4, the identified master regulators have different number of links with the target genes whether we are in tumour or normal conditions. For instance, a closer look at *ARNT2*, revealed a regulatory relationship with 30 genes in both conditions and 152 specific to tumour. Differences in out-degree could potentially explain why the conserved MR$^2$ do not always show the lowest SEG−index. Integrating out-degree metric in the mathematical definition of housekeeping genes and dissecting what makes these regulatory modules condition-specific, using for example gene set enrichment analysis (GSEA) [60], could be an interesting future investigation with several potential implications. Further, given the involvement of MR in tissue development and their well-known roles in some clinical diseases [61], we find that the extensive research effort surrounding the identification and characterization of MR by computational methods could gain additional

insight by integrating conditional dependence (*i.e.* the proposed $MR^2$ procedure) as pruning step in their respective algorithms.

## Conclusion

We proposed two novel approaches that cast the GRN reconstruction problem as a blend between regularized multivariate regression and graphical models. Through extensive comparative analysis with simulated and real-world data, we demonstrated that the introduced models are consistent and exhibit excellent performance over the contenders. Considering the often encountered dilemma in GRN inference whereby a choice has to be made between linear and non-linear modeling assumptions, we further show that consideration of multiple responses even in a linear setting can show as good performance as non-linear approaches (e.g. random forests). In addition, without assuming any prior on TFs nor inferring them from the individual models built for the target genes, the $L_1L_{2,1}$ and $L_2L_{2,1}$ leverage sparsity in the regression coefficients and precision matrix to identify master regulators while offering the possibility to infer their plasticity and regulatory interactions. Future research in this area will be directed towards consideration of time-delay effects in the proposed models as well as designing efficient techniques for hyperparameters tuning that account for the imbalanced nature of gold standard networks often encountered in GRN inference.

## Materials and methods

### Data sets

**DREAM5.**   To evaluate the performance of the proposed and contending approaches, we used three benchmark data sets from the DREAM5 challenge freely available from [20]. As summarized in Table 5, each data set contains a collection of gene expression profiles, a gold standard (*i.e.* a set of verified interactions) and a list of known TFs. Briefly, network 1 is a simulated data set mimicking the transcriptional regulatory network of *E. coli* in which 10% of random edges were added and the expression profile generated with GeneNetWeaver [62]. For network 3 and network 4, the Gene Expression Omnibus (GEO) database [63] was used to produce affymetrix genuine gene expression data sets for *E. coli* and *S. cerevisiae* respectively. The resulting microarray data sets where then normalized using Robust Multichip Averaging (RMA) [64]. For a detailed description of the DREAM5 inference challenge, its design and the data generation process, interested readers are referred to [20] and the DREAM website.

**E. coli time-resolved transcriptomics data.**   The ability of the proposed methods to reconstruct GRN with small sample data across multiple conditions or tissues is evaluated by further considering time-resolved transcriptomics data resulting from the experiment in [65], available from the GEO database under accession GSE20305. Here, we investigate the gene expression responses of *E. coli* strain MG1655 to four stress conditions (*i.e.* oxidative stress,

**Table 5. Details of gene expression data sets for model organisms *E. coli*, *S. cerevisiae*, as well as *in silico* from DREAM5.**

| Networks | #Samples | #TFs | #Genes | #Verified interactions |
|---|---|---|---|---|
| *In silico* (Network 1) | 805 | 195 | 1643 | 4012 |
| *E. coli* (Network 3) | 805 | 334 | 4511 | 2066 |
| *S. cerevisiae* (Network 4) | 536 | 333 | 5950 | 3940 |

For each network, this includes the number of putative TFs, TGs, samples and verified interactions in the gold standard. The original labels of each network from the challenge are given in parentheses.

glucose-lactose diauxic shift, heat, and cold). Except for the scenario where stress was induced by hydrogen peroxide (*i.e.* oxidative stress), sampling with 10 min steps for transcript profiling was performed from time points 10–50 min post-perturbation plus two control time points prior to each perturbation. Averaging over the three available biological replicates for each time point resulted to the expression profile data of five samples for individual stress condition and 4400 genes.

**Human lymphoblastoid cell lines.** Using the gold standard given by the functional regulatory network built from the intersection of functional and binding edges in [66], the proposed approaches were further validated on two expression data sets for natural variation from human lymphoblastoid cell lines (LCL) from [67] and [68] available respectively from GEO accession GSE23120 and EBI ArrayExpress accession E-GEUV-3. These are referred to as Niu and Geuvadis respectively. Considering only genes present in the expression profile lead to a gold standard with 17 TFs, 2755 target genes and all together 6389 verified interactions.

**Transcriptome data set for non-small cell lung cancer.** To further investigate the identification of master regulators across different conditions, we employ the expression profiles of 10077 genes from ten independent GEO data sets with a total of 1118 non-small cell lung cancer (NSCLC) samples including both primary tumours (925 samples) and tumour-free control (193 samples) lung tissues. The data has been reprocessed (*i.e.* merged, normalized, batch effect-corrected and filtered for genes with low variance across samples) using a robust statistical methodology and the tumour samples were curated to include only primary NSCLC (*i.e.* squamous cell carcinoma (SCC), adenocarcinoma and large cell carcinoma (LCC). Detailed information along with the preprocessing steps can be found in [69, 70]. It is worth pointing that, the pipeline and data freely made available by the authors are of capital importance for further downstream analysis, whereby the limited accessibility of such large-scale genomic data to people without a proper background in bioinformatics and the time consuming preprocessing step often required are overcame. For performance assessment, we used as gold standard the pancancer regulon from DoRothEA [71], that is a collection of TFs and their transcriptional targets curated and collected from different types of evidence for both human and mouse. Since DoRothEA assigns five different confidence levels ranging from A (highest) to E (lowest) between interactions, we considered levels A to D interactions and selected only those with TFs (*i.e.* from human) present in the latest version of the transcriptional regulatory relationships unraveled by sentence-based text mining (TRRUST) [72], a manually curated database of human and mouse transcriptional regulatory networks. Further preprocessing the expression profile to account only for genes present in the ground truth lead to a final data set with 5490 genes of which 625 were TFs.

## Data pre-processing, hyperparameter tuning and evaluation metrics

As a pre-processing step, the expression levels of each gene are centered and scaled within each data set. To tune hyperparameters $\lambda_1$ and $\lambda_2$, we used 10-fold cross-validation (CV) and split each gene expression profile data set from DREAM5 into 10 non-overlapping subsets of almost identical size. With $s_1 = \left\{ \frac{\gamma}{10} : \gamma = 1, \cdots, 20 \right\}$ and $s_2 = \{2^{-\delta} : \delta = 0, \cdots, 8\}$ as the search spaces for $\lambda_1$ and $\lambda_2$ respectively, we finally select the optimal $\lambda_1$ and $\lambda_2$ as the maximizer of the log-likelihood on the validation data. Due to the very small sample size in the case of time-resolved data sets, leave-one-out CV was used instead with the same grids. Interestingly, we observed that model performance is more influenced by $\lambda_2$, the penalty on the regression coefficient matrix. We further found that there is a limiting factor for which irrespective of the chosen $\lambda_1$, $\Omega$ results in a diagonal matrix. This is very useful for the practical implementation as it

can be used to efficiently reduce computation time while controlling the amount of sparsity in the precision matrix.

Regarding performance evaluation, we follow the DREAM5 strategy and only consider the top 100,000 edge predictions to evaluate TF-TG interactions as a binary classification problem for which, edges are predicted to be present or absent. With the selected interactions, we then make use of area under the receiver operating characteristic (AUROC) and area under the precision-recall (AUPR) curves, two widely used metrics for performance assessment in GRN inference. For an overview of the performances across all used data sets, we also computed the score for each metric and the overall score as shown in Eq (17).

$$
\begin{cases}
\mathrm{AUROC_{score}} &= \left( \prod_{i=1}^{n} \mathrm{AUROC}_i \right)^{\frac{1}{n}} \\[2em]
\mathrm{AUPR_{score}} &= \left( \prod_{i=1}^{n} \mathrm{AUPR}_i \right)^{\frac{1}{n}} \\[2em]
\mathrm{Overall_{score}} &= \dfrac{\mathrm{AUROC_{score}} + \mathrm{AUPR_{score}}}{2}
\end{cases}
\tag{17}
$$

where $n$ is the number of considered networks (*e.g.* in the current analysis, $n = 3$ and $n = 5$ for respectively DREAM5 and time-resolved transcritomics data sets).

Because of the imbalanced property of ground truth networks in GRN inference, using AUPR and AUROC to compare models with different level of sparsity may not be ideal. For instance, a false positive edge may be penalized even if it doesn't exist in the gold standard. In addition, precision and recall at a given threshold $k$ may not consider the ranking of each edge [73]. As a result, two networks could have the same number of true and false edges at threshold $k$, resulting in the same precision and recall values but with a different ranking for the considered edges. For these reasons, further performance assessment was conducted using early precision (EP) [74] (*i.e.* the fraction of true positives in the top-$k$ edges excluding self-loop) and normalized discounted cumulative gain (nCDG) [73, 75] computed for every edge in the true positive set of the gold standard network and defined in Eq (18).

$$
\begin{cases}
\mathrm{nDCG_{network,}}_k = \dfrac{\mathrm{DCG_{network,}}_k}{\mathrm{IDCG_{gold\ standard,}}_k} \\[2em]
\mathrm{DCG_{network,}}_k = \displaystyle\sum_{i=1}^{k} \dfrac{x}{\log_2(i+1)} \\[2em]
\mathrm{IDCG_{gold\ standard,}}_k = \displaystyle\sum_{i=1}^{k} \dfrac{1}{\log_2(i+1)} \\[2em]
x = \begin{cases} 1 & \text{if edge is true positive} \\ 0 & \text{if edge is false positive} \end{cases}
\end{cases}
\tag{18}
$$

where k is the number of true positive values in the gold standard network.

In addition, recalling that for the proposed approaches we would like to quantify the contribution of individual TF on the remaining genes (*i.e.* respectively rows and columns of our estimated regression coefficient matrices), we scale TF-wise, edge weights obtained from each inference method to range in the interval [0, 1]. That is, for the $i^{\mathrm{th}}$ row $\boldsymbol{\beta}^i = [\beta_1, \cdots, \beta_s]$ of the

estimated coefficient matrix, the maximum absolute scaling is used to compute each normalized entry as $\frac{|\beta_j|}{\max|\boldsymbol{\beta}^i|}$.

## Contending approaches

To provide a comprehensive comparative analysis, we compared the solutions of the proposed models with nine state-of-the-art approaches. To account for updated developments in GRN inference and because our analysis relies on the data sets from DREAM5 challenge, we selected D3GRN [76], PLSNET [77], ENNET [78], PORTIA and its extension etePORTIA [41] as some of the most recent state-of-the-art approaches that used the same data sets. Further, we included those methods that were ranked among the top three GRN reconstruction approaches in the challenge based on the overall score. These approaches included: TIGRESS [17], that was deemed the best linear regression-based method in DREAM5, GENIE3 [79], that uses variable selection with ensembles of regression trees and ANOVerence [80] that relies on the non-linear Cohen's correlation coefficient $\eta^2$ computed from two-way analysis of variance (ANOVA). We also included the Fused LASSO [14] formulation that combines information from multiple data sets, shown to outperform contending approaches.

## Identification of master TFs

The term "master regulator" refers to a TF that is at the top of the transcriptome regulatory hierarchy, thus regulating the majority of other TFs and associated TGs [81]. Using the common paradigm in GRN inference, whereby it is assumed that a TF-TG edge is causally oriented from TF to TG, and that the set of TG includes TF, we used the estimated sparse regression coefficient and precision matrix from the proposed models to identify the master regulator type 1 and type 2 (i.e. $\text{MR}^1$ and $\text{MR}^2$). Given the estimated sparse regression coefficient matrix $\widehat{\mathbf{B}} \in \mathbb{R}^{p \times s}$ and precision matrix $\widehat{\boldsymbol{\Omega}} \in \mathbb{R}^{s \times s}$, we say that a TF (i.e. column of the predictor matrix $\mathbf{X} \in \mathbb{R}^{n \times p}$) is a type 1, $\alpha$–master regulator ($\text{MR}^1_\alpha$) if for $0 < \alpha \leq 1$, the corresponding row in $\widehat{\mathbf{B}}$ has an $\alpha$–percentage of non-zero entries. For example, let us assume that a row vector for a given TF (e.g. TF1) contains 80 non-zero entries out of 122 associated TGs. From this, we obtain $\alpha = 0.65$ (i.e. 80/122), and we say that TF1 is a $\text{MR}^1_{65}$. That is, about 65% of the corresponding TGs are found associated with TF1. Regarding type 2 master regulator ($\text{MR}^2_\alpha$), we used conditional dependence (i.e. non-zero TF-TG entries in the sparse precision matrix) to validate that the same TF-TG in $\widehat{\mathbf{B}}$ is non-zero. While enhancing the sparsity in the regression coefficient matrix, this procedure also serves to validate if the direct link identified by $\widehat{\mathbf{B}}$ remains a link given the rest of genes in the network. Finally, similar to type 1, $\widehat{\mathbf{B}}$ derived from this procedure is then used to detect what we call $\text{MR}^2_\alpha$. Without loss of generality and for ease of notation, the subscript $\alpha$ will be dropped throughout the text unless specified otherwise.

## Supporting information

**S1 Fig. Enrichment analysis of conserved $\text{MR}^1$ and $\text{MR}^2$ with time-resolved transcriptomic data sets from *E. coli*.** Shown are the fold enrichment sorted by GO biological process. (A) $\text{MR}^1$ found conserved across the four stress conditions. (B) $\text{MR}^2$ conserved under cold, lactose and oxidative stress. We used the graphical gene-set enrichment tool "ShinyGO" v.0.76.1 http://bioinformatics.sdstate.edu/go/ for the analysis.
(EPS)

**S1 Table. Comparison of model performance using area under the ROC curve (AUROC) and area under the precision-recall curve (AUPR) on DREAM5 data sets.** The reported results are from the DREAM5 challenge and correspond to the best (i.e. overall score) inference methods that participated in the challenge. Since results obtained using the R package "precrec" were slightly different from those of the challenge (cf. Table 1), we sought to include the latter here to have a comprehensive assessment and to avoid misinterpretation of the current results.
(PDF)

**S2 Table. Specificity index for MR$^1$ & MR$^2$ across cold, heat, lactose and oxidative conditions.** Using the proposed $L_1L_{2,1}$ and $L_2L_{2,1}$, we derived a list of master transcription factors genes (i.e. MR$^1$ & MR$^2$) conserved in the four stress conditions. The $\tau$-index shows the condition-specificity of each gene in each condition.
(XLSX)

**S1 Text. Supplementary methods.** The Text includes detailed explanations on how to derive: (1) The matrix of regression coefficients B, as the solution to a special case of Sylvester equation, (2) The special cases of the $L_1L_{2,1}$ and $L_2L_{2,1}$ solutions as well as the precision matrix $\Omega$ as the solution to a special form of algebraic Riccati equation.
(PDF)

## Author Contributions

**Conceptualization:** Alain J. Mbebi, Zoran Nikoloski.

**Data curation:** Alain J. Mbebi.

**Formal analysis:** Alain J. Mbebi.

**Funding acquisition:** Zoran Nikoloski.

**Investigation:** Alain J. Mbebi, Zoran Nikoloski.

**Methodology:** Alain J. Mbebi, Zoran Nikoloski.

**Software:** Alain J. Mbebi.

**Supervision:** Zoran Nikoloski.

**Validation:** Alain J. Mbebi, Zoran Nikoloski.

**Visualization:** Alain J. Mbebi.

**Writing – original draft:** Alain J. Mbebi, Zoran Nikoloski.

**Writing – review & editing:** Alain J. Mbebi, Zoran Nikoloski.

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
