## [Decision Letter · Decision Letter 0]

1 Mar 2023

Dear Dr. Nikoloski,

Thank you very much for submitting your manuscript "Gene regulatory network inference using mixed-norms regularized multivariate model with covariance selection" for consideration at PLOS Computational Biology.

As with all papers reviewed by the journal, your manuscript was reviewed by members of the editorial board and by several independent reviewers. In light of the reviews (below this email), we would like to invite the resubmission of a significantly-revised version that takes into account the reviewers' comments.

We cannot make any decision about publication until we have seen the revised manuscript and your response to the reviewers' comments. Your revised manuscript is also likely to be sent to reviewers for further evaluation.

Sincerely,

Miguel Rocha

Guest Editor

PLOS Computational Biology

Kiran Patil

Section Editor

PLOS Computational Biology

Reviewer's Responses to Questions

**Comments to the Authors:**

Reviewer #1: Gene regulatory networks are the underlying mechanism of controlling the expression of genes. Although lots of inference methods have been proposed to solve the problem, most of them ignore to analyze the characteristic of regulators and target genes. In this paper, the authors proposed two novel GRN reconstruction approaches to incorporate the L2,1-norm penalty with regularized multivariate regression and covariance selection. The experimental results on DREAM5 datasets show the good performance of the proposed method. However, there are some questions I concerned:

1) The application of L2,1-norm in reconstruction of gene regulatory networks is not a novel idea (similar idea in the following citation), so the authors need to compare them and clarity the main contribution.

Gui S, et al. BMC bioinformatics, 2017, 18: 1-13.

2) The authors should discuss the effects of time information on the construction of gene regulatory network, such as the time-delay effect on regulation.

3) As single-cell transcriptome becomes popular in recent years, if the proposed method is also effective in scRNA-seq data?

4) The outputs of the proposed method are sparse networks, while the outputs of GENIE3 are the full networks, so the comparison of AUROC and AUPR are not so fair about the tail of ordered regulation relationships. The authors could use the early precision ratio instead.

Pratapa A, et al. Nature methods, 2020, 17(2): 147-154.

5) Although the proposed method achieves the comparative performance than other non-linear ones (like GENIE3), the analysis of these two kinds of methods should be performed, such as the intersections of the results or advantages of these methods.

6) Some recent citations are missing, such as :

[1] Moerman T, et al. Bioinformatics, 2019, 35(12): 2159-2161.

[2] Zheng R, et al. Bioinformatics, 2019, 35(11): 1893-1900.

[3] Shu H, et al. Nature Computational Science, 2021, 1(7): 491-501.

Reviewer #2: The paper "Gene regulatory network inference using mixed-norms regularized multivariate model with covariance selection" by Mbebi and Nikoloski tries to quantify the power of joint methods (regularize multivariate regression and graphical models) in the classic task of reverse engineering gene regulatory networks (GRNs). In order to do so, the authors use data from DREAM5. Below, my points.

- An issue of the paper is that it uses data from two monocellular organisms, E.coli and S.cerevisiae. The task of GRN reconstruction has notoriously a step-up in obstacles when performed in higher eukaryotes. I suggest therefore that the authors apply their promising method in a higher eukaryote. The simplest task that comes to mind would be choosing a large dataset (e.g. a TCGA lung cancer dataset, which is publicly available and providing ~450 tumor samples plus 50 normal control samples). The optimal method for network reconstruction and master regulator definition (in the contrast tumor vs. normal) should be able to identify cancer genes (e.g. taken from the golden set at COSMIC Cancer Gene Census) with a higher precision (or accuracy) than simpler non-blended methods. Any other higher eukaryote dataset (e.g. a vertebrate, or even Drosophila) would be sufficient, the rationale behind this request being that E.coli and S.cerevisiae reconstruction is easier and has far less importance, due to the near-completion of regulatory networks in these species (e.g. in the RgulonDB project).

- The MR algorithms used by the authors use the common paradigm in GRN reconstruction literature, which briefly assumes that a TF-target edge is causally oriented from the TF to the target gene. It would be better if the authors clearly stated this causality assumption. Also, given the evident experience of the authors, they could discuss if their proposed inference approaches may benefit from causal algorithms to improve MR definition. This would improve the role of this paper in the development of a future generation of causal inference methods in Master Regulator Analysis.

- The article is very technical and is hard to read for a biological audience. There is a general lack of clarity when presenting the performance of the proposed method vs. other methods, and an almost non-existent explanation of how the Master Regulator is evaluated. I would suggest presenting the results in a more widely intuitive way, for example clearly stating, in the main text tables, precision and accuracy of the proposed inference, based on different parameters of -for instance - the L2L2,1 norms models.

- Connected to the previous point, the supplementary figures should definitely be moved to the main text (which currently has NO figure). This should be associated with a graphical improvement of the figures themselves. For example, Figure S1 should have higher resolution, thicker lines, and more clear color definition (for example with different combinations of colors and line types.

- I am glad the authors provided their code on Github, however the code is currently irreproducible, as the authors didn't provide some of the raw data required to execute it. One example is in the file L1L21_Dream5_Scerevisia_example_run.R which at line 7 requires the non-existing file "net4_expression_data.tsv", and at line 10 "net4_transcription_factors.tsv". Until the code is made fully reproducible, including the generation of the figures in the text, I cannot critically evaluate its correctness.

Reviewer #3: The authors presented a new method for gene regulatory network (GRN) inference based on sparse linear regression and covariance matrix estimation. Four different models have been proposed, and benchmarked against state-of-art GRN inference techniques on the well-known DREAM5 data set. Because the gold standard networks from DREAM5 are mostly based on simulations, the authors also had to complement their work with real-world networks. In particular, they used time series transcriptomic data from a previous study on E. Coli, sampled across different tissues and experimental conditions. In terms of GRN reconstruction, the proposed method shows encouraging performance compared to already existing methods, especially given the simplicity and elegance of the model. Finally, they demonstrated how their approach can be used for identifying master regulators based on the sparsity patterns of the inferred weight matrices. The main contribution of this work mainly lies in the mathematical developments, which all prove to be correct. The manuscript is overall well-written and easy to read.

Major comments

--------------

Four inference approaches have been proposed, namely $L_1 L_{2, 1}$, $L_1 L_{2, 1}G$, $L_2 L_{2, 1}$ and $L_2 L_{2, 1}G$. Based on the text, it is pretty clear that $L_1 L_{2, 1}$ and $L_2 L_{2, 1}$ differ by the regularisation function used to penalise the precision matrix $\\Omega$. However, I could not find what $G$ refers to, and what the 2 additional methods $L_1 L_{2, 1}G$ and $L_2 L_{2, 1}G$ consist in.

Table 2 is suffering from severe shortcomings, as some of the methods present in table 1 are missing in table 2. The 4 proposed methods indeed seem to outperform Fused Lasso on the time-resolved transcriptomics data sets and Fused Lasso has already been applied on the same data in the past, but the publication dates back from 2016 and has never been compared to PORTIA, ENNET and PLSNET, among others. The authors should include all the contenders in table 2.

While linear modelling of the relationship between transcription factors (TF) and target genes (TG) remains highly relevant (the use of non-linear models is not well motivated in the literature actually), the choice of using $L_{2, 1}$ regularisation still lacks motivation. The authors assumed that the number of TFs is considerably smaller than the number of TGs, and that each TF is likely to regulate many TGs. On synthetic data sets, this design choice might lead to poorer performance. For example, in DREAM3 and DREAM4 in silico networks, most genes regulate at least one TG each, and each regulates a few of them. Moreover, even in real-world settings, this can happen as a side-effect of gene pre-selection (e.g. based on a priori knowledge or quality-based filtering). Therefore, it is crucial that the authors clearly demonstrate the relevance of $L_{2, 1}$ regularisation, either by (1) further validating their models on additional data sets for which the gold standard networks do not have this idealised topology, or (2) illustrating through some basic graph-theoretical analysis or literature review that most gold standard networks actually reflect such row-wise sparsity patterns. Optionally, the authors could also comment on these limitations and provide recommendations on how to manually tune the hyper-parameter $\\lambda_2$. Ideally, a script could be provided on the github repository for the automated tuning of $\\lambda_2$ based on 10-fold cross-validation.

The authors reported that "all MR1 that the proposed L1L21 and L2L21 found conserved across all four conditions 231

have a maximum condition specificity index of .533". Is the inverse true as well? Are the genes with low specificity index listed among the genes selected by the models in all four conditions? Currently, table 3 does not give a good sense of the specificity and sensitivity of this approach. A more informative figure showing the distributions of all tau indexes would be valuable. Also, it is not clear from the text that there are only 3 MR1 TFs and 3 MR2 TFs (as suggested by the table). If more master regulators have been inferred by the proposed methods, I would recommend showing the whole list.

Also, the presence of 0 tau indexes for genes alaS and flhC is surprising. In the original paper where the tau index presented in equation 17 was introduced, the expression values $x_i$ are supposed to be normalised by the maximum. This should be mentionned in the text for clarity purposes, because it is unclear whether the maximum value has been computed across genes or samples. Assuming that the authors normalised across genes, then based on equation 17 this would mean that alaS and flhC both had the highest expression value, and these values were the same in all experimental settings. This is extremely unlikely, so I assume the authors normalised across samples. However, this still means that these 2 genes had constant expression, which is unlikely as well. Could the authors comment on that? Overall, I think table 3 and the corresponding section should be revised in order to clarify all these points.

While the manuscript is backed by an open-source implementation of each of the 4 models on the github repository, I would recommend adding standalone scripts to produce the results shown in tables 1 and 2. In particular, I did not find the code necessary to run their models on the time-resolved E. Coli data sets for GRN reconstruction and estimation of the tau indexes.

Given the high level of technical details present in the paper, by contrast the solution presented in equation 9 is lacking context and developments. It is the only equation that cannot be deduced just by reading the text and without relying on external ressources. I would recommend adding more developments to it in suppl. mat.

Minor comments

--------------

Type-2 master regulators have been defined based on the sparsity of the weight matrix and the "non-zero TF-TG entries in the sparse precision matrix". Since the precision matrix is of shape TG x TG, the authors are implicitely assuming that the set of TGs necessarily contains the set of TFs. This could be made more explicit in the text.

The Sylvester equation (eq. 8) is not homogeneous, due to the independent term $X^T Y$ being different from 0. Same remark for suppl. eq. S1.

There are minor tense inconsistencies, like "We use the estimated sparse regression coefficient" or "we use three benchmark data sets from the DREAM5", when most of the manuscript is written in the past tense.

"numner of TGs" -> "number of TGs"

Conclusion

----------

Based on the simplicity, elegance, but also the novelty of the proposed approaches and the good performance shown on the DREAM5 benchmark, I would recommend making major revisions to the current manuscript. All the contenders listed in table 1 should be present in table 2 to allow a fair comparison. Table 3 and section about master regulator identification should be deeply revised to make it clear that the agreement between inferred master regulators and tau indexes is not due to random chance. Finally, the authors should investigate the relevance of their modelling choices (e.g. $L_{2, 1}$ regularisation) in other settings than the ones tested so far (e.g. DREAM5), and discuss how the topology of gold standard GRNs can affect the inference accuracy.

**Have the authors made all data and (if applicable) computational code underlying the findings in their manuscript fully available?**

Reviewer #1: Yes

Reviewer #2: Yes

Reviewer #3: Yes

PLOS authors have the option to publish the peer review history of their article (what does this mean?). If published, this will include your full peer review and any attached files.

Reviewer #1: No

Reviewer #2: No

Reviewer #3: No
---

## [Decision Letter · Decision Letter 1]

11 Jul 2023

Dear Dr. Nikoloski,

We are pleased to inform you that your manuscript 'Gene regulatory network inference using mixed-norms regularized multivariate model with covariance selection' has been provisionally accepted for publication in PLOS Computational Biology.

Best regards,

Miguel Rocha

Guest Editor

PLOS Computational Biology

Kiran Patil

Section Editor

PLOS Computational Biology

I would recommend the comments from reviewer 3 to be taken into account in the final version of the manuscript.

Reviewer's Responses to Questions

**Comments to the Authors:**

Reviewer #1: The authors have addressed all my questions

Reviewer #2: The authors replied satisfactorily to my comments. I particularly thank them for the update to the Github page, as reproducibility is one of the most important results of any computational study.

Reviewer #3: All of my previous concerns have been carefully addressed by the authors. I would like to mention the extra effort put in the benchmarking, the comprehensiveness of the results shown on the E. Coli and LCL datasets, and the diversity of the evaluation metrics used. My new comments are the following:

In table 2, it appears that early precision is subject to strong variability that does not seem to be of high significance. Could the authors comment on this? A striking example is the 99.7% early precision of L2L1G on the LCL dataset (Niu), while L2L1 (a very similar method) produces 64.75%. Furthermore, 99.7% seems very high especially given the very poor AUROC (51.8%). It would be worth reporting the number of (true) positives, as well as investigating the reason of such variability across methods.

Highlighting in table 3 is very misleading due to numerous errors (assuming bold text corresponds to the best contenders). As an example, PORTIA has not been highlighted for "Cold AUPR". For "Heat AUROC", PORTIA should be the best contender (0.517 > 0.515). Examples are plentiful. I acknowledge that this task is tedious due to the table size, and do not assume that these mistakes were done on purpose.

Was the TF-wise normalization performed for all the performance metrics (AUROC, EP, etc.)? If that is the case, this post-processing choice should be substantially motivated, as many of the contending approaches try to address the comparability of their predicted scores between TFs/TGs (e.g., variance scaling in ENNET, row-column weighting in PORTIA). For this reason, normalizing by the maximum might penalize methods that explicitly correct for this comparability issue. I would suggest the authors to check the impact of removing that normalization step to make sure results are consistent.

**Have the authors made all data and (if applicable) computational code underlying the findings in their manuscript fully available?**

Reviewer #1: None

Reviewer #2: Yes

Reviewer #3: Yes

PLOS authors have the option to publish the peer review history of their article (what does this mean?). If published, this will include your full peer review and any attached files.

Reviewer #1: No

Reviewer #2: No

Reviewer #3: No

---

## [Editor Report · Acceptance letter]

27 Jul 2023

PCOMPBIOL-D-22-01880R1 

Gene regulatory network inference using mixed-norms regularized multivariate model with covariance selection

Dear Dr Nikoloski,

I am pleased to inform you that your manuscript has been formally accepted for publication in PLOS Computational Biology. Your manuscript is now with our production department and you will be notified of the publication date in due course.

With kind regards,

Anita Estes
